# Potential for Natural Fiber Reinforcement in PLA Polymer Filaments for Fused Deposition Modeling (FDM) Additive Manufacturing: A Review

**DOI:** 10.3390/polym13091407

**Published:** 2021-04-27

**Authors:** Ching Hao Lee, Farah Nadia Binti Mohammad Padzil, Seng Hua Lee, Zuriyati Mohamed Asa’ari Ainun, Luqman Chuah Abdullah

**Affiliations:** 1Laboratory of Biocomposites, Institute of Tropical Forestry and Forest Products (INTROP), Universiti Putra Malaysia, UPM Serdang, Selangor 43400, Malaysia; 2Laboratory of Biopolymer and Derivatives, Institute of Tropical Forestry and Forest Products (INTROP), Universiti Putra Malaysia, UPM Serdang, Selangor 43400, Malaysia; 3Faculty of Engineering, Universiti Putra Malaysia, UPM Serdang, Selangor 43400, Malaysia

**Keywords:** additive manufacturing, natural fiber-reinforced polymer composite filaments, fused deposition modeling, kenaf fiber, 3D printing technology

## Abstract

In this review, the potential of natural fiber and kenaf fiber (KF) reinforced PLA composite filament for fused deposition modeling (FDM) 3D-printing technology is highlighted. Additive manufacturing is a material-processing method in which the addition of materials layer by layer creates a three-dimensional object. Unfortunately, it still cannot compete with conventional manufacturing processes, and instead serves as an economically effective tool for small-batch or high-variety product production. Being preformed of composite filaments makes it easiest to print using an FDM 3D printer without or with minimum alteration to the hardware parts. On the other hand, natural fiber-reinforced polymer composite filaments have gained great attention in the market. However, uneven printing, clogging, and the inhomogeneous distribution of the fiber-matrix remain the main challenges. At the same time, kenaf fibers are one of the most popular reinforcements in polymer composites. Although they have a good record on strength reinforcement, with low cost and light weight, kenaf fiber reinforcement PLA filament is still seldom seen in previous studies. Therefore, this review serves to promote kenaf fiber in PLA composite filaments for FDM 3D printing. To promote the use of natural fiber-reinforced polymer composite in AM, eight challenges must be solved and carried out. Moreover, some concerns arise to achieve long-term sustainability and market acceptability of KF/PLA composite filaments.

## 1. Introduction

Additive manufacturing (AM), generally known as 3D printing, is a material-processing method of adding materials layer by layer to create a three-dimensional object. Polymer, metal, and ceramic can be applied in 3D printing. To avoid confusion, important terms are highlighted in Figure 1. AM is a term to describe a manufacturing process, 3D printing is the technology that performs AM, and a 3D printer is the processing tool. Before AM was introduced, subtractive manufacturing was applied in most manufacturing industries. A bigger piece of raw material is subtracted (scraped, dissolved, turned, and/or bored) to produce a specific shape of product [1]. Therefore, a lot of waste is produced, even though it is a faster processing method. AM achieves minimum waste scrap production, as the material adds up to the pinpoint location for every layer, making waste management insignificant. It is also an economically effective tool for small-batch production. There is high flexibility regarding product geometry, which can be varied from time to time or even from piece to piece without altering processing tools. To date, AM has emerged in many manufacturing divisions, including aerospace, and civil, biomedical, and automotive engineering [2]. Countless achievements have been marked during the history of the technology. The global 3D printing market is estimated to grow by approximately 17.00% CAGR and is expected to gain USD 4390 million by 2027 [3].

Unfortunately, current 3D printing still cannot compete with conventional manufacturing processes. It is hardly employed in a mass production industry that produces thousands of parts every day. Small and/or customized design products are preferred, as the mold cost of an injection molding machine costs tens of thousands US dollars. Nevertheless, the extremely high material efficiency of AM is needed in the near future along with the fourth industrial revolution (IR 4.0). 3D printing can be mainly categorized into filament-, liquid-, or powder resin-based AM. Among the types of 3D printing processes, fused deposition modeling (FDM), stereo lithography (SLA), and selective laser sintering (SLS) are the most frequently used for these three different printing principles, and they will grow continuously (Figure 2). Of all the techniques, FDM is the most popular working principle for embedding natural fiber reinforcement [4]. Preformed composite filaments made it easy to print using a 3D printer without or with minimum alterations to hardware parts. Moreover, PLA polymer is most frequently used, due to its relative lower melting temperature and straight polymer chains.

Natural fiber-reinforced polymer composite (NFRPC) filaments are receiving great attention because it is economical and highly biodegradable. Unfortunately, challenges to the homogenous distribution of natural fibers in the matrix are frequently highlighted. Non-homogenous composite flow in the 3D printer nozzle may cause uneven printing and/or a clogging situation (stuck nozzle) [6]. Powder- and liquid-based 3D printers may have more difficulties because the lighter-density natural fibers always “float” to the top of the matrix [7]. Moreover, void production has been observed, mainly for two reasons: (1) low interfacial bonding in the composite due to the incompatibility between hydrophilic natural fibers and the hydrophobic matrix; and (2) the nature of FDM 3D printing technology, which is unable to join layers in a way in which the bottom layer is solidified before the layer above is stacked on top. Furthermore, environmental conditions and process and production parameters are reported to influence the performance of the product [8]. Nevertheless, natural fiber reinforcement improves the material’s mechanical properties. Natural fiber promotes a better load-transfer mechanism in the composite.

Kenaf fiber is one of the best-known plant fibers for reinforcement in polymer composites. In Malaysia, it has been planted with an intention to replace tobacco as well as to prevent wood-logging. The kenaf plant was selected due to its fast growth rate, superior reinforcement properties, and high photosynthesis rate. The excellent performance of kenaf fibers and its promising reinforcement ability can be easily tracked in the literature [9]. Specific strength and stiffness properties comparable to glass fibers can be observed [10]. It provides strength reinforcement for PLA polymer composites and creates composite products low in cost and weight. It also acts as a nucleating agent and reduces glass transition temperature, making it higher in flexibility.

The use of natural fiber-reinforced polymer composite in AM is still in its infancy. However, there is no denying that it has gained a lot of attention and favor in both industry and academia in recent years. Despite its bright future, there are eight challenges that need to be overcome or addressed, mentioned by Fidan et al. [11]. At the same time, some future scope of works must be undertaken to develop fiber-reinforced polymer composites for FDM 3D printing technology. In recent years, there have been considerable achievements made to strength enhancement for 3D-printed polymer composites for FDM 3D printing. However, 3D printing with kenaf fiber reinforcement is yet to be observed. However, the kenaf fiber has a good track record for composite property enhancement, along with other natural fibers. Therefore, we discuss and review the potential of kenaf fiber polymer composite filaments for FDM 3D printing. At the beginning of this review, the FDM 3D printing technique will be reviewed. This will be followed by an introduction to the kenaf plant fiber. After that, we explore and discuss the potential for kenaf fiber reinforcement in FDM 3D printing. Challenges and future perspectives for kenaf fiber reinforcements in AM are included in final section. This review serves to promote the use of kenaf fiber in PLA composite filaments for FDM 3D printing.

## 2. Additive Manufacturing (AM): Fused Deposit Modeling (FDM) 3D Printing Technology

The working principle of the FDM method is relatively simple compared to other 3D-printing methods. FDM, or fused filament fabrication (FFF), is an additive manufacturing process that belongs to the material extrusion family. Figure 3 shows the standard setup of an FDM 3D printer. In FDM, a continuous thermoplastic polymer filament is heated above its melting temperature in a nozzle. This molten polymer will fall, due to gravity, on a printing bed through the nozzle, which is traveling according to code generated by CAM software. The molten polymer quickly solidifies at room temperature and forms a layer of polymer. An object is built by selectively depositing melted material in a predetermined path, layer by layer, to create a 3D structure. The first layer of printing is the most crucial in determining the success of the product. This non-laser application is the main reason that FDM 3D printing is relatively cheap and easier to maintain.

A poor printing surface is always an issue for FDM printing. The molten polymer drops onto a flat bed and solidifies via thermal conduction and convection. The shrinkage of the previous layer causes structural deformation, known as warping or curling [12]. This is cause by poor bonding between the two adjacent layers that solidify at different times. However, this issue can be regulated by lower cooling rates or smaller temperature gradients [13]. Incomplete fusion between layers leads to weak mechanical properties (Figure 4). Both interlayer and in-layer failure modes have been discovered during FDM printing, leaving printed products with low tensile strength. Therefore, applications of the PLA filament are limited to display modeling only [14].

To generate good bonding, modeling analysis shows higher extrusion and environmental temperature facilitates reasonable bonding on polymer segments. Additionally, Cummings (2016) installed four piezoelectric transducers on the flatbed of a FDM machine to inspect real-time conditions of the partially formed model in every 30 s, to detect small geometric defects in the printed product [15]. It showed that good correlation between filament feeding rate, wall geometry, and G-code-defined wall structure produces high-quality 3D-printed products [16]. This shows that a lot of work is required to improve the printing precision of the FDM technique.

Furthermore, consumable filaments generally have low melting points, and well-developed polymers such as polylactic acid (PLA), acrylonitrile butadiene styrene (ABS), and polyamide (PA) are among favorite choices. The materials are the cheapest among all types of additive manufacturing, and PLA filament is the most frequently used due to its low cost, relatively lower melting temperature, and straight polymer chains. Bardot and Schulz (2020) reviewed the PLA filaments used in FDM 3D printing [18]. The PLA filament melt process starts at 190–200 °C, and a lower temperature difference, compared to room temperature, reduces the severity of cold shrinkage. A high shrinkage rate of the printed product reduces its dimensional precision, and this is more significant for a higher temperature gradient and stress during FDM printing.

To break this constraint, scientists have made various reinforcements to polymer filaments. The use of fiber-reinforced polymer composite filaments has offered higher strength performance. Long fiber reinforcements require modification to the printing head (extruder). A continuous carbon-fiber insertion has recorded triple and 150% performance improvements in tensile and flexural properties [19]. Delamination, matrix cracking, and fiber-matrix debonding and fiber breakage are the main causes of failure, showing room for improvement by repairing the fiber-matrix interface. A pressure roller is then developed to apply pressure along the “fresh” printed path. The study shows a positive trend to the strength improvement of up to 0.5 mm pressing level, due to enhanced interface bonding [20]. However, short carbon-fiber reinforcements may not observe as much strength improvement. Insufficient fiber length for effective interface bonding and high porosity deteriorates tensile strength [21,22]. Chemically treated carbon-fiber PLA filament increases its tensile strength by 12%. Furthermore, natural fiber-reinforced PLA filaments continuously receive attention due to their biodegradability.

## 3. Development of Natural Fiber-Reinforced PLA Polymer Composite for FDM 3D Printing

In AM, natural fibers known as renewable resources such as hemp, kenaf, flax, jute, and more can be applied as reinforcing agents. These sources are impregnated with many types of polymers that perform differently in terms of characteristics and properties. Mazzanti et al., (2019) coined the term additive bio-manufacturing (ABM), which involves extra factors such as cell growth, bioactivity, and service life [23]. This technique is mainly dedicated to biomedical applications via FDM. Researchers are finding that most acceptable methodologies or techniques offer long-lasting constancy and performance in terms of stronger and improvised fiber-matrix bonding. In fact, most concerns regard the obtaining of an environmentally friendly product or process by exploring raw materials and composites made from NFRPC [24]. Table 1 shows enhancement developments and difficulties encountered in previous studies.

There are certain requirements that NFRPC needs to fulfill to be processed by AM, namely (1) types of reinforcements and matrices; (2) good fiber-matrix bonding; (3) fiber homogeneity; (4) fiber alignment; (5) good interlayer bonding; and (6) minimal porosity. Fiber reinforcement of appropriate size, shape, and length needs to be selected to suit the intended purpose of the part. Both matrix material, which holds the fibers in place, and reinforcement need to be compatible with the selected 3D printing technique. A good fiber-matrix bond is required at the fiber-matrix interface to allow loads to be transferred efficiently from the matrix, thus resulting in composites that follow the “rule of mixture.” Fiber-loading is also crucial for obtaining AM composites with good mechanical properties. Homogeneity in fiber distribution is needed to ensure consistent properties throughout the printed part. The added ability to control fiber distribution and alignment in a predefined location and direction allows the strengthening of sections of an object. A good interlayer fusion is required to avoid delamination. Finally, unwanted voids that would affect the mechanical properties of NFRPC should be minimized [25].

Emphasis is paid to FDM 3D printing techniques that focus on continuous fibers, as they hold potential to become next-generation composite fabrication methodologies. However, a special heating nozzle must be designed to fit long fiber-reinforced polymer composites. Short-fiber polymer filaments are extruded via the extruder and then fed into the commercial heating nozzle for printing. This simple printing process has prompted many studies focusing on short-fiber reinforcement. However, limited studies have been found regarding natural fibers with polymer mixtures in powder form for sintering/binding technology. This is because natural fibers and polymers are not spread evenly in each layer due to different densities. Fonseca Coelho et al., (2019) pointed out that the use of gypsum sisal fiber powder in binder jetting 3D printing method disturbs the next layer of material deposition, and some of the sisal fiber diameter is thicker than the layer thickness, producing a higher amount of porosity and deteriorating the strength [26]. Fortunately, post-processing in this study induced a good fiber-matrix interface and enhanced sample strength.

In one study, twisted yarns of jute fibers were used to reinforce PLA matrix using fused filament fabrication [27]. These composites exhibited tensile strength and modulus of 57.1 MPa and 5.11 GPa, which are 134% and 157% higher than that of pure PLA, respectively. However, the tensile strain was very low, between 0.05 and 0.25%. The authors indicated that the appropriate pre-tensioning of jute yarn fibers can help in achieving uniform molding and improved mechanical properties. Therefore, Hinchcliffe et al., (2016) attempted to fabricate jute and flax fiber (continuous) reinforced PLA composites using AM. They found that pre-stressing the continuous fiber can improve tensile and flexural properties [28]. Pil et al., (2016) discussed the use of flax and hemp fibers which are popular as reinforcement in polymer composites. Besides their environmentally friendly values, these natural fibers aided the composites to become stiffer than glass-fiber composites in both tension and plate bending [29]. In fiber-oriented composites, flax and hemp fibers are beneficial for reinforcing agents in AM. For example, a fiber arrangement for a 3D printing process for designed furniture, especially desk lamps, can be very impactful.

As mentioned by Le Duigou et al., (2016), a study was carried out to understand the printing parameters dedicated to mechanical properties of FDM in AM [30]. They discovered a kind of biocomposite called hygromorphic biocomposite. The biocomposite contained natural fibers to produce original apparatus that was able to actuate within a moisture gradient. The researchers demonstrated that wood fiber-reinforced biocomposite FDM provided stronger mechanical properties based on 0 or 90° of printing orientation. The commercially available FFF filament, consisting of PLA þ poly(hydroxyalkanoate) (PHA) matrix, was reinforced with 15.2 wt.% recycled wood fibers to create composite samples. The properties of these composites were found to be comparable to conventionally processed (extrusion and IM) PP-30% wood and HDPE-40% wood composites, but lower than PHA-20% wood composites. The sample size (print width) was found to have a strong influence on the total porosity of the samples, and therefore their mechanical properties. For example, as shown in Figure 5a,b, as-received composite filament had ~ 16.5% porosity, which resulted in a similar amount of porosity (14.7–15.5%) to the FFF processed samples (Figure 5c). However, the porosity increased with sample size (print width) due to the loss of deposit temperature when printing large samples with longer deposition paths (Figure 5d), which resulted in a decrease in the properties. The samples with short deposition paths resulted in a considerable reduction in porosity (8.4–14%). The thermal consolidation of the filament and printed samples resulted in a significant drop in porosity of these samples, as shown in Figure 5e. They also found that thermal consolidation can reduce moisture absorption by these composites. From these preliminary studies it can be said that research on AM of NFRPC is in its embryonic stage, and a significant amount of research is required to understand and optimize the process.

Hinchcliffe et al., (2016) demonstrated the enhancement of PLA via (1) additive manufacturing; and (2) continuous natural fiber (such as jute and flax) [28]. The type of fibers, matrix geometry, reinforcing strands, and degree of post-tension in continuous natural fibers of PLA were also investigated using tensile and flexural testing. They claimed that AM individually can improve tensile and flexural mechanical properties by obtaining 116%, 62%, 14%, and 10% of tensile specific strength, stiffness, flexural specific strength, and rigidity accordingly. In another study by Milosevic et al., (2017), natural fiber-reinforced pre-consumer recycled polypropylene composites were prepared for FDM [32]. The natural fibers chosen were hemp or harakeke fibers. Comparison was made between reinforced composites and plain polypropylene samples. The reinforced composites or filaments showed tremendous results of tensile strength and Young’s modulus by having more than 50% and 143% increment than plain samples. An amount of 30 wt.% hemp or hanakeke (*Phormium tenax*) was found to provide optimum results.

Stoof and Pickering (2017) applied harakeke, one of New Zealand’s distinctive native plants, as a successful filament in AM innovations [33]. The natural fibers were found to be a potential value-adding material to the polymer in terms of increasing its mechanical and aesthetic properties. They investigated filaments which contained polypropylene (PP), and 30 wt.% harakeke exhibited good results of tensile strength and Young’s Modulus of 41 MPa (77% increment) and 3.8 GPa (275% increment) accordingly. In another study, Stoof et al., (2017) focused on the application of hemp and harakeke in the making of the NFRC dedicated to AM [34]. Natural fibers of hemp and harakeke were dispersed with a PLA polymer to produce 3-mm uniform filaments. Results showed that hanakeke would be a useful natural fiber, which enhanced 42.3% and 5.4% of Young’s modulus and tensile strength, respectively, compared to plain PLA. To aid environmental aspects, Horta et al., (2018) developed composites from recycled thermoplastics and natural fibers of wood residues via AM (FDM) [35]. Therefore, a list of parameters has been studied, such as effect of temperature, distance between layers, deposition speed, and dimensional accuracy.

The natural fibers that are categorized as short-fiber-reinforced filaments is applied. However, the application of such fibers is still limited. Two studies conducted by Ning et al., (2015) and Ning et al., (2017) focused on the use of carbon fiber at various concentrations and lengths, which showed better results in terms of tensile and flexural properties [36,37]. Ivey et al., (2017) reported that tensile properties—mainly the elastic modulus of the MEAM samples—increased by adding carbon fibers (CF) to the polylactic acid (PLA) filament making [38]. Therefore, the filament of PLA/CB was better than PLA alone.

According to Sunny et al., (2017), the use of short natural plant fiber (SPNF) can limit problems of agglomeration and dispersion, because such fibers can align better [39]. In AM, fiber orientation is a crucial factor that has a major influence on the composite performance during either dry or wet processing. Tekinalp et al., (2019) hypothesized and proved that nanocellulose that is found beneficial in plant natural fibers provides significant mechanical property improvements in the making of AM polymers [40]. However, the fibril morphology, dispersion, and adhesion aspects need to be investigated. The nanocellulose, which is cellulose nanofibrils, was dispersed with polylactate matrix, which resulted in an increment of more than 80% tensile strength and 200% elastic modulus. In short, Terkinalp et al., (2019) successfully demonstrated the preparation of full biobased renewable feedstock material for AM.

**Table 1 polymers-13-01407-t001:** Enhancement development and difficulties for natural fiber-reinforced PLA composite filament in FDM 3D printing.

Type of Natural Fiber	Enhancement Development	Challenges	Reference
Short Hemp and Harakeke fiber	Tensile Strength improvement	Sample surface appeared less glossy and coarser. Pores forming due to insufficient fusion of the layers during printing process	[34]
Short Hemp Fiber	Storage modulus and Elastic modulus enhancement	A reduction of viscosity due to pectin degradation during melt mixing stage	[41]
Short Bamboo Fiber	Reinforcement in thermal and strength properties achieved by adding nanoclay	High brittle of specimen due to the nanoclay insertion Degradation of polymer found during mixing stage	[42]
Long flax yarn	4.5 times of tensile strength and modulus enhancement	The filaments produced in non-perfect circular shape due to immature fabrication process. Poor impregnation of yarn filament creates porosity and microstructures. Overlapping printing found at the corner spot. Low performance at transverse direction	[30]
Commercial Wood	Better printing formability than ceramic-, aluminum- and copper-based PLA filament	Weak interlayer bonding was observed contributed to low strength performance	[21]
Wood short fiber	Initial deformation resistance has enhanced	Poor interfacial bonding between wood fiber and PLA polymer reduced the strength values. Lower thermal stability wood fiber reduced the overall thermal performance	[43]
Wood flour	Better interfacial bonding with coupling agent	Unflavored color changes on the filament Rougher fracture surface with localized plastic deformations. Crystalline hindrance in filament produces more amorphous regions. Higher water uptake Print nozzle was clogged due to agglomeration of wood flour	[44]
Silk fiber	Retain antibacterial properties for scaffold printing	-	[45]

## 4. Kenaf Plant Fibers

Three decades ago, Williams (1990) mentioned that natural fibers have a tendency to improve biological characteristics in developing natural composites for biomedical purposes [46]. Natural fibers have often been classified based on their origin such as animal, mineral, or plant fibers. These fibers have different chemical constituents, with animal fibers containing a higher percentage of protein, and plant fibers having more cellulose. Plant fiber constitutes about 60–80% cellulose, 5–20% hemicellulose and lignin, and about 20% waxes, moisture, and pectin, as well as water soluble organic components [31]. The agricultural residue of plant fibers can be derived from many parts of the plant, including bast, leaf, fruit or seed, grass, or reed. Therefore, their cellulose and cellulose crystallinity also vary accordingly. The same goes for the physical properties of plant fibers that have different densities ranging from 1.1 to 1.6 g/cm^3^ [31]. The advantages of applying plant fibers can be listed as: (1) the potential to focus on functional, complex, and net-shaped parts; (2) the ability to manufacture site-specifically with functionality; and (3) the possibility to produce tailored fiber alignment [31].

Kenaf fiber (*Hibiscus cannabinus* L.) is in the bast fiber family, which is selected due to its high strength [47]. The properties of kenaf fiber are caused by its chemical composite compositions, cellulose, hemicellulose, and lignin constituents. Each of the components provides insignificant different functionality, and therefore shows slightly different properties to natural fibers. The dependence of different properties of natural fibers on their constituents, summarized in Figure 6, shows contradicting compositional requirements to achieve desired properties such as mechanical, thermal, biological, and moisture absorption. High concentration of hemicellulose in the fiber provides high thermal stability to natural fibers during processing and use. However, high hemicellulose in the natural fiber is detrimental to its mechanical properties and consequently to composite properties. Similarly, high crystalline cellulose is beneficial for improved kenaf fiber mechanical properties with a concomitant decrease in biological degradation and moisture resistance. It is interesting to note that most properties can be improved if lignin can be restricted to low concentration. It is believed that fiber treatments such as alkaline treatment (mercerization) are very effective in removing lignin, which not only improves fiber-matrix interactions and boding but also enhances the other properties depicted. It has been discussed that the addition of natural fibers to thermoplastics and thermoset plastics offers several benefits, such a low cost, eco-friendliness, renewability and lower damage to processing equipment compared to synthetic reinforcements.

Kenaf fiber has been one of the raw resources for fiber-based industries and paper production in Malaysia since 2000. The kenaf plantation project in Malaysia was initiated in 1999 by the National Economic Action Council (NEAC). It formed a steering committee to study the potential to grow kenaf in Malaysia as another industrial crop, and was intended to replace tobacco, and prevent wood-logging for the paper industry. The kenaf plant was selected due to its fast growth, high productivity/land ratio, extraordinary photosynthesis rate, and suitability in the Malaysian climate. However, Paridah et al., conducted a cost-benefit analysis on kenaf fiber production in Malaysia and it found not so profitable. Kenaf plantation should be aided by government, giving consideration to kenaf plantations’ contribution to socioeconomic and environmental benefits [48].

Kenaf fibers are one of the most well-known plant fibers used for reinforcement in polymer composites. It is an herbaceous annual plant, which can be grown under a wide range of weather conditions with 1–10 cm/day growth rate [49]. The large difference in plant heights is due to various factors such as cultivation method, plantation season, field light intensity, plantation period, intensity, and age of the plant cultivars. Rouison (2004) discovered that kenaf has been used in two distinct manners [50]. Kenaf takes up nitrogen and phosphorus in the soil. Applying 90 kg N/ha of nitrogen was reported to improve plant growth significantly [51]. These minerals also contributed to increasing the overall weight of weeds, increasing the height of crops, the diameter of stems, and the yield of fiber. Moreover, kenaf has a significantly higher photosynthesis rate when compared to grass or conventional trees. As compared to conventional trees under 1000 μmol/cm^2^/s, the photosynthesis rate of kenaf is 23.4 mg CO_2_/dm^2^/h. This shows that kenaf plants are environmentally friendly, not only due to their biodegradability, but also because they release more oxygen and reduce carbon dioxide than other plantations.

Conventional NFRPC has been manufactured through techniques such as manual layup, resin transfer molding, spray-up, automated tape laying, automated fiber placement, filament winding, and pultrusion [25]. One common issue with the majority of conventional techniques is the need for molds, which not only makes processing more expensive, but also limits the formability of final part. As a result, producing complex and customized parts becomes tedious and costly. The need for low-cost design flexibility and an automated fabrication process has spurred the development of AM for NFRPC. AM has brought about a revolution in the way we can manufacture complex products with customized features. AM has paved the way for application areas ranging from aerospace to automotive to consumer to biomedical. Composites produced by AM have attracted special attention due to their promise in improving, modifying, and diversifying the properties of generic material by introducing reinforcements.

Multiple studies of kenaf reinforcement have been carried on thermoplastic [52,53] and thermoset [54] polymer composites, due to its superior strength of single-fiber analysis [55]. Furthermore, lots of kenaf fiber polymer composite review papers have been published, showing its high interest regarding fiber reinforcements [56,57,58,59]. Its biodegradability reduces some major issues such as disposal, non-renewability, expense, non-biodegradability, and other environmental problems. However, despite being known as renewable, abundantly available, biodegradable, environmentally friendly, cost-effective, and light-weight, as well as portrayed as having strong future potential and prospects, the properties of kenaf fiber-reinforced PLA polymer composites still require improvements to be comparable to synthetic fiber-reinforced polymer composites and to reach long-term sustainable performance and stability [31].

### 4.1. Potential Use of Kenaf Fibers for PLA Polymer Composite in FDM 3D Printing

Although many natural fibers have already been adopted in polymer composite filaments for FDM 3D printing, kenaf fiber reinforcement has not yet been seen in previous studies for 3D printing. Kenaf fiber-reinforced PLA (KF/PLA) composite is one of the most famous natural fiber polymer composite combinations, and has undergone intense research in recent decades. A KF/PLA composite was studied and found to have similar strength enhancements to other natural bast fiber PLA composites [60]. Interfacial bonding at the fiber-matrix boundary is the main factor for determining strength. An excess of fiber contents leads to insufficient resin for complete fiber wetting. Optimum kenaf fiber contents were found at a 30 wt.% loading [61]. A large variation in fiber dimension made it deviates from theoretical/modeling analysis. Higher fiber volume fraction recorded bigger differences between experimental and theoretical values [62]. This is because high fiber contents cause an insufficient amount of resin for fiber wetting, resulting in bad interface adhesion and thereby disturbing the load-transfer mechanism from matrix to the fiber [63]. Poor fiber-matrix ratio also resulted in extreme high–low viscosity. Both conditions may lead to a high number of void contents, which act as stress concentration spots, receiving lower load capacity before cracking occurs. Besides that, the removal of non-cellulosic components on the fiber surface helped to improve interfacial bonding, and hence strength performance [64]. Alkaline treatment is cost-effective, and a relatively simpler fiber surface treatment than reported in previous study. It exposes reactive sites to cellulose components and provides mechanical interlocking via rough-surface topography, to ensure better bonding with the PLA matrix [65].

On the other hand, kenaf fibers act as a nucleating agent and reduce glass transition temperature, making them higher in flexibility. Moreover, they provide another reinforcing mechanism and encourage the nucleation of less common β-phase crystals to achieve higher impact strength and toughness [66]. Similar effects were reported for hemp [67], jute [68], and coir [69] reinforced PLA composites. The abovementioned studies shows that kenaf fiber has very high potential for use in PLA filaments as other natural fiber PLA composites.

Today, plant-fiber reinforced PLA composites have been developed and fabricated into filament shapes for use in FDM 3D printing. PLA composite filaments are claimed to have better strength performance. Therefore, KF/PLA composite filament should also join this innovation application due to its well-tracked property enhancements by kenaf fiber reinforcement. However, some challenges face scientists to apply kenaf fiber reinforcement in PLA composite filaments for 3D printing. This will be discussed in the following section, with some future perspectives.

### 4.2. Comparison between Synthetic and Natural Fibers as Reinforcing Agent

Synthetic fibers have already found space in the field of additive manufacturing earlier than natural fibers. A review by Goh et al., (2019) mainly emphasized synthetic fibers as a reinforcing agent. Generally, the tensile strength of the composites reinforced with synthetic fibers reportedly increased by more than 100% compared to pure polymer counterpart [25]. According to a review by Sekar et al., (2019), various synthetic fibers were used as a reinforcing agent, e.g., short glass fiber, short carbon fiber, and continuous carbon fiber. All the studies reported increases in modulus and tensile strength in the reinforced composites [8].

By comparison, the adoption of natural fiber-reinforced composites in 3D printing technology is relatively new. Moreover, the properties of the resultant composite did not increase drastically, particularly in terms of mechanical properties, compared to that of pure polymers. Stoof and Pickering (2017) initiated a study using harakeke and hemp fibers as reinforcing agents for PLA composite [34]. The results revealed that the surface of the samples became coarser, and voids appeared throughout the samples. These voids could act as stressing concentration points and lead to samples failing, which consequently lead to inferior mechanical properties. Poor adhesion was also observed, as indicated by fiber pull-out. On the other hand, tensile strength was decreased along with increasing fiber content. Contrarily, Young’s modulus increased with increasing fiber content. However, optimum results were obtained when 20 wt.% harakeke fibers were used, where an increment of 5.4% and 42.5% in tensile strength and Young’s modulus, respectively, was recorded. Nevertheless, this improvement was not worth mentioning when compared to synthetic fibers. Matsuzaki et al., (2016) reinforced PLA with continuous carbon fiber and found an improvement of 335% in tensile strength compared to pure polymer with merely 6.6 vol.% fiber-loading [27]. Despite that, the issue can be resolved by optimizing some processing parameters.

## 5. Challenges and Future Perspectives

Natural fiber-reinforced polymer composite in AM is still in its infancy. However, no one could deny that it has gained a lot of attention and favor in both industry and academia in the recent years. Starting in 2014, the numbers of publications related to the topic “fiber—composite—AM” or “carbon fiber—composite—3D printing” has increased drastically from around 10 publications in 2014 to around 160 publications in 2019 [70]. It is interesting to note that the accumulated papers on the topic was around just 20 publications before 2014. There has been a gratifying growth in the number of publications over the course of 6 years, which indicates the tremendous potential of AM.

AM has a very promising future in the manufacture of natural fiber-reinforced polymer composites. According to Ho et al., (2012), conventional processing methods used in producing natural fiber-reinforced polymer composites are compression molding, resin transfer molding, injection molding, hot pressing, and vacuum infusion molding [71]. However, in competing with the conventional methods, several universities and research institutes have already started to adopt fiber-reinforced additive manufacturing (FRAM) techniques to print lightweight and sustainable components. In fact, car frames made from carbon-fiber-reinforced acrylonitrile butadiene styrene (ABS) has already been produced using AM, as a replacement to serial production, by several researchers from Oak Ridge National Laboratory [72].

Despite its bright future, there are some challenges that need to be overcome or improved by the current FRAM technique. It is anticipated that these challenges will frame the directions of future research on the topic. The main challenges are summarized in Figure 7.

Reviews done by Goh et al., (2019), Fidan et al., (2019), and Zindani and Kumar (2019) have identified some future scope for works regarding FDM 3D printing technology of fiber-reinforced polymer composite [11,25,73]. This future work includes: (i) material development; (ii) fiber-matrix interfacial properties; (iii) fiber homogeneity; (iv) fiber alignment; (v) interlayer bonding; (vi) porosity; and (vii) printability. Mazzanti et al., (2019) also mentioned problems of natural fiber reinforcement in FDM printing [23]. Figure 8 display the concept map of factors critical to the fabrication of composites using AM techniques.

Technological challenges in the processing of continuous fiber and fiber sheets have resulted in more than 80% of studies on FDM composite materials using short fibers as a reinforcing agent. Nevertheless, the demand for stronger materials for AM will call for more applications of continuous fibers in the future. Apart from that, the future challenge in AM comes from the quantification of the interface properties of the composite. Currently, most interface properties are characterized qualitatively using scanning electron microscope (SEM) images. More detailed research is therefore required to evaluate the interfacial strength of composites. Fiber homogeneity and fiber alignment are also important factors that need to be considered in future works, as it will affect the transfer efficiency of stress from the matrix to the reinforcements.

Regarding interlayer bonding, the bonding between adjacent layers is very important, because FDM is a layer-by-layer process. Tensile properties in the z-direction were found to be weaker compared to those in-plane [74]. It was reported that the ultimate tensile strength in the z-direction is only two thirds that in the x- and y-directions. Van Hooreweder et al., (2013) reported that interlayer fracture is more brittle than intralayer fracture, as lower fracture strain in the z-direction has been observed compared to the x-direction [75]. Improving interlayer bonding of filament materials in future work would be beneficial when producing materials with better mechanical properties.

Porosity refers to the number of pores or voids that exist in materials. These unfilled spaces are undesirable, as they are bound to negatively affect the material integrity and subsequently lead to poor mechanical properties of the materials [70]. The cause of pores or voids is therefore vital so that the right steps can be taken to eliminate or reduce these pores. The porosity is affected by printing parameters such as laser power, beam speed, and scan spacing [76]. In addition, the incorporation of natural fibers also affects the formation of porosity due to incompatibility in the fiber-matrix interface [77]. Therefore, future work should focus on the optimization of the printing parameters and fiber pre-treatment to enhance the compatibility of the fiber-matrix interface. Another issue when dealing with natural fibers as reinforcing agents is the alteration of the rheology properties of the materials, which affects the printability of the materials [25]. Controlling rheology response such as the addition of a rheology modifier could help to solve this issue.

Moreover, NFRC filaments suffer from low mechanical properties due to fiber-matrix incompatibility and an inherently weaker nature of natural fibers compared to that of synthetic fibers [78]. These issues can be effectively addressed by hybridization with (i) synthetic fibers and polymer matrices in addition to natural fibers; or (ii) more than one type of natural fibers with different mechanical and other characteristics/properties. Other considerations for long-term sustainability and market acceptability of KF/PLA filaments include (i) continuous and reliable sources of kenaf fibers; (ii) energy and cost involved in the kenaf fiber supply chain; (iii) tax incentives for production, use, and recycling of kenaf fiber and its filament (Nuunna et al., 2012). Variations in kenaf fiber supply during seasonal and unseasonal times can affect its quantity, cost, quality, and characteristics, and therefore the production and properties. As discussed above, the hybridization of composites using different natural fibers with kenaf fiber can also address deficiencies in fiber supplies. Establishing processing and storage facilities near farmland can provide significant cost benefits in the production of kenaf fiber-reinforced PLA composite filaments.

## 6. Conclusions

Additive manufacturing (AM), generally known as 3D printing, is a material-processing method of adding materials layer by layer to create a three-dimensional object. AM achieves minimum waste production, making waste management insignificant. Unfortunately, current 3D printing still cannot complete with conventional manufacturing processes due to its low production rate. It is an economically effective tool for small-batch or high-variety product production.

Of all techniques, FDM is the most popular working principle for embedding natural fiber-reinforced polymer composites. Being preformed of composite filaments makes it easy to print using a 3D printer without or with minimum alteration to hardware parts. A continuous filament of a thermoplastic polymer is heated to above its melting temperature in a nozzle. An object is built by selectively depositing melted material in a predetermined path, layer by layer, to create a 3D structure.

Unfortunately, 3D-printed products with pure polymer exhibit low performance, limiting its range of applications. Prototype models are usually printed for display purposes, but are hardly ever treated as functional models. Therefore, natural fiber-reinforced polymer composite (NFRPC) filaments are attracting attention because they are economical and highly biodegradable. Uneven printing, clogging, and inhomogeneous distribution of the fiber-matrix interface are the frequently highlighted challenges. Void observation of FDM 3D-printed products was proof of low interfacial bonding and poor joining between layers. Nevertheless, natural fiber reinforcement did improve mechanical properties, paving the way for functional products.

Kenaf fibers are one of the best-known plant fibers for reinforcement in polymer composites. They are available abundantly, biodegradable, and are one of the famous natural fibers in the history of NFRPC. Kenaf provides strength reinforcement for PLA polymer composites and creates composite products low in cost and weight. It also acts as a nucleating agent and reduces glass transition temperature, making it higher in flexibility.

Natural fiber-reinforced polymer composite in AM is still in its infancy. Eight challenges and some future work must be solved to develop fiber-reinforced polymer composites for FDM 3D printing technology. Moreover, some researchers have mentioned the need to achieve long-term sustainability and market acceptability of KF/PLA composite filaments. Variations in kenaf fiber supply during seasonal and unseasonal times can affect their quantity, cost, quality, and characteristics, and therefore its production and properties. Establishing processing and storage facilities near farmland can provide significant cost benefits in the production of KF/PLA composite filaments.

Although kenaf fiber has a good track record regarding composite property enhancement as well as with other natural fibers, 3D printing with kenaf fiber reinforcement has not yet been seen in previous study. We think that kenaf fiber has much potential and should be brought up to a higher stage. Therefore, this review serves to promote kenaf fiber in PLA composite filaments for FDM 3D printing.

## Figures and Tables

**Figure 1 polymers-13-01407-f001:**
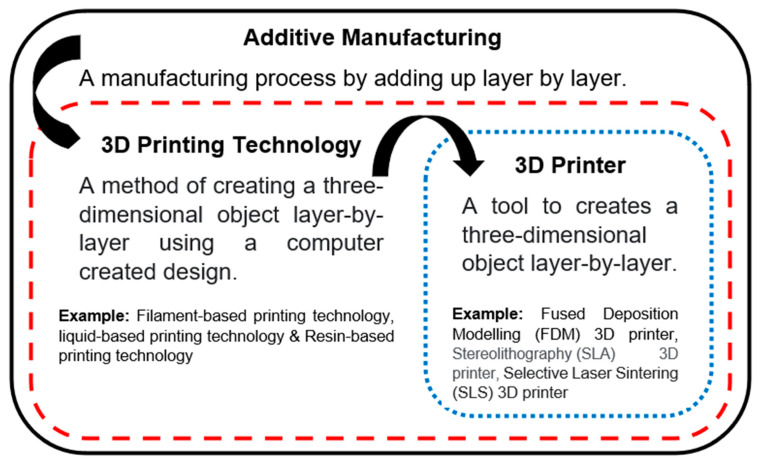
Important terms in additive manufacturing.

**Figure 2 polymers-13-01407-f002:**
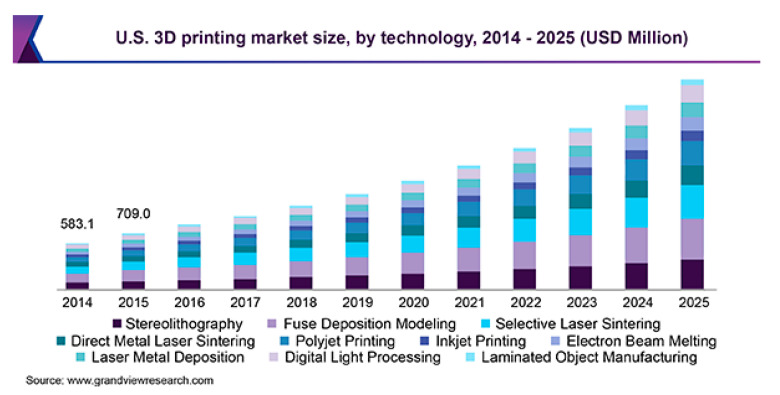
3D printing market size forecasting in the U.S., sorted by 3D printing technology, estimated based on 2017 [5].

**Figure 3 polymers-13-01407-f003:**
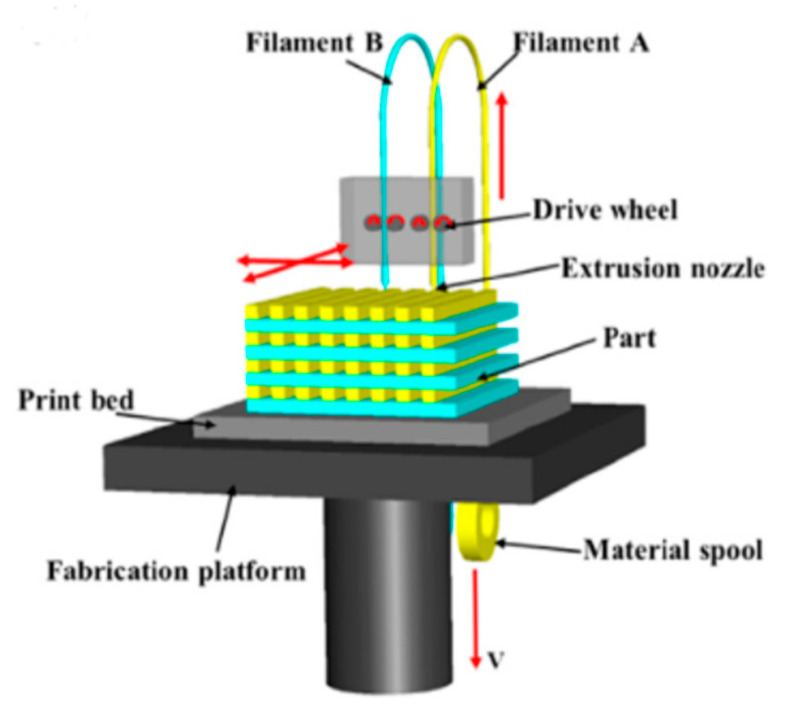
Schematic of the FDM printing method [4].

**Figure 4 polymers-13-01407-f004:**
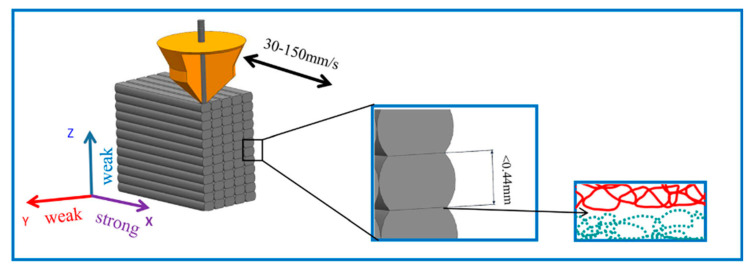
Incomplete fusion between layers and weak mechanical properties [17].

**Figure 5 polymers-13-01407-f005:**
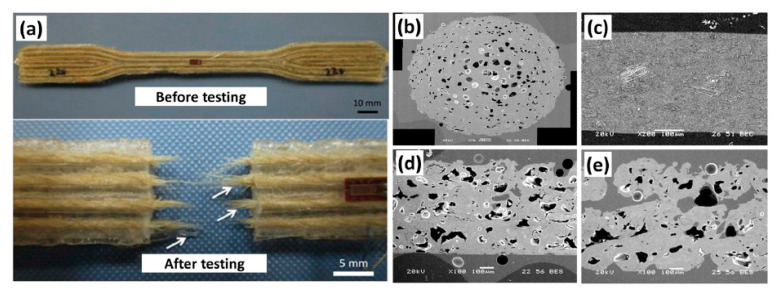
(**a**) Typical tensile test specimen of FFF processed jute fiber-reinforced PLA composites (top) and fiber pullouts after testing (below); Microstructures of PLA–PHA–wood fiber composites (**b**) as-received filament, (**c**) consolidated FFF sample, (**d**) 100% print width, 300% print width [31].

**Figure 6 polymers-13-01407-f006:**
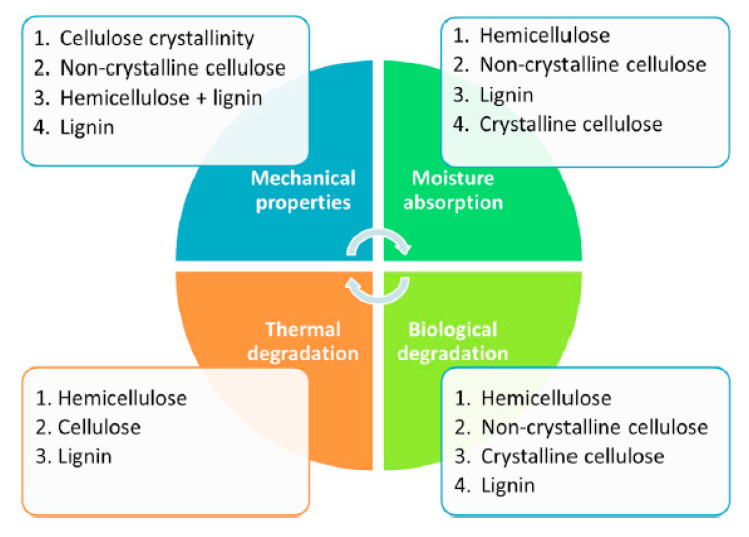
Contribution of chemical constituents of natural fibers on selected properties.

**Figure 7 polymers-13-01407-f007:**
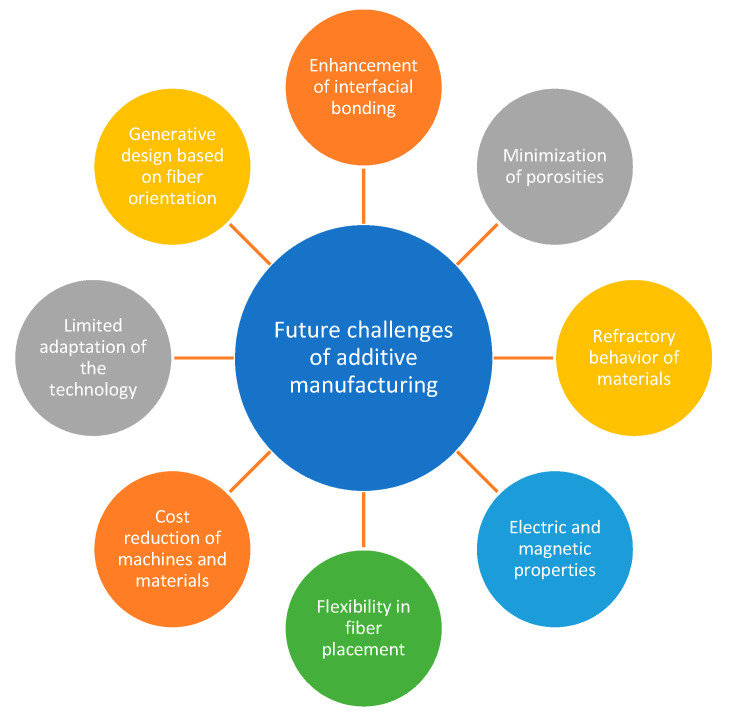
Future challenges of natural fiber-reinforced polymer composites in AM (modified from [11]).

**Figure 8 polymers-13-01407-f008:**
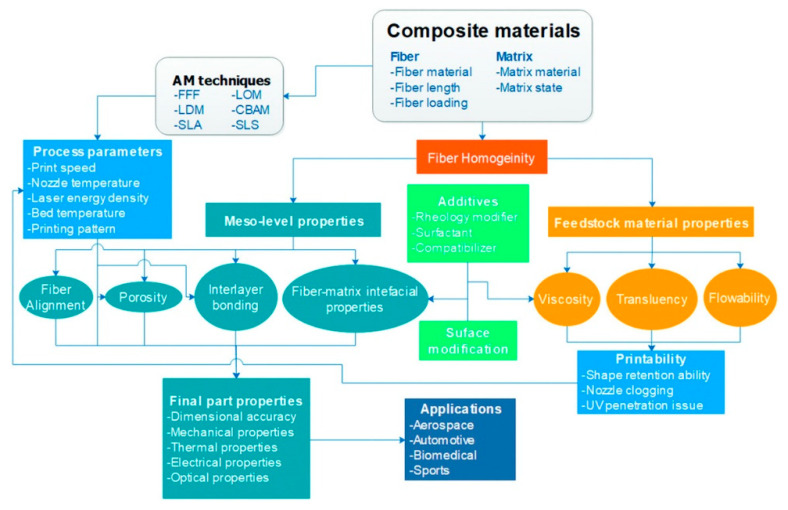
Concept map of factors critical to the fabrication of composites using AM techniques [25].

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
