# Peer review of "Potential for Natural Fiber Reinforcement in PLA Polymer Filaments for Fused Deposition Modeling (FDM) Additive Manufacturing: A Review"

_polymers, 2021, doi:10.3390/polym13091407_

Round 1

Reviewer 1 Report

The English language needs to be extensively checked with a language specialist, as there are phrases that have no meaning, strange word usage and lots of grammatical errors.

The first 7 pages of the paper are actually a review of AM and therefore these sections (1 and 2) should be compressed to a minimum, as this is not really the subject at hand.

Please do not use terms like "recently" when referring to work already reported in 2016 (page 9). Or terms like "little study" (also on page 9).

Is Figure 7 own authors work? Or is it taken from reference [40] ? Do you then have permission to use it?

The same applies for Figure 9 and 10.

From the title of the paper one may deducted that the review is only of Kenaf Reinforced PLA 3D printed parts, but in reality just a small sectin is on that. I strongly suggest to change the title of the paper accordingly (e.g NFRC  Reinforcements in PLA .....).

In a review, some should also include a comparison with other reinforcements (for example glass fiber) and their results/improvments.

Author Response

The English language needs to be extensively checked with a language specialist, as there are phrases that have no meaning, strange word usage and lots of grammatical errors.

  • The manuscript has sent to English proofreading.

The first 7 pages of the paper are actually a review of AM and therefore these sections (1 and 2) should be compressed to a minimum, as this is not really the subject at hand.

  • The paragraphs involving AM has compressed and shorten.

Please do not use terms like "recently" when referring to work already reported in 2016 (page 9). Or terms like "little study" (also on page 9).

  • The improper words were removed and substituted by other words.

Is Figure 7 own authors work? Or is it taken from reference [40] ? Do you then have permission to use it?

The same applies for Figure 9 and 10.

  • The reuse permission required for tables and figures in our manuscript was obtained again, since multiple revisions has been done. The permission files were attached together in this correction.

From the title of the paper one may deduct that the review is only of Kenaf Reinforced PLA 3D printed parts, but in reality just a small section is on that. I strongly suggest to change the title of the paper accordingly (e.g NFRC Reinforcements in PLA .....).

  • Thank you for your valuable suggestion. This is because few natural fibers have been studied in 3D printing filament yet none of them using kenaf fibers. However, our goal for this review is to promote the potential of kenaf fiber in PLA composite 3D printing filament. Hence this is the main reason why kenaf PLA 3D printed composite section is only in minor, but it is representing our main objective. I hope this clarifies our intention.

In a review, some should also include a comparison with other reinforcements (for example glass fiber) and their results/improvments.

  • After our discussion, we decided not to insert any comparison with synthetic reinforcements. The reason behind this is because synthetic fillers enhance composite properties better than natural fibers. The synthetic comparison section may cause the readers to loss their interests in the use of kenaf reinforcement. Hence, we only provide reviews on other natural fiber reinforcements. I hope this clarifies our objective in this review paper. 

Reviewer 2 Report

”..high variety product production”: what does the authors mean?

“Hence, natural fiber reinforced polymer composite filaments seekhigh19attentionin the market.”: There is no connection to the last sentence and this one. So, why does author use “hence”?

The language used and the writing style are poor. Authors have to improve the language and the writing style should be more academic/research.

There are articles published earlier on 3D printing of kenaf based filaments unlike authors says in the text.

Figure 2 directly copied from the source. Did authors take permission? It is common to adapt and redraw in author’s own way. The same for figure 3, 4, 5 …

References are not marked as per journal rules

Only PLA polymer is mentioned. Isn’t there any other polymer that has been used?

Is table 2 even necessary for this review? Any relevance?

Kindly see the below articles for further references

1) FDM 3D Printing of Polymers Containing Natural Fillers: A Review of their Mechanical Properties

2) Additive manufacturing (3D printing): A review of materials, methods, applications and challenges

3) A review of 3D and 4D printing of natural fibre biocomposites

4) 3D printing using plant-derived cellulose and its derivatives: A review

The current version has to be improved technically; and also the language must be improved

Author Response

”..high variety product production”: what does the authors mean?

  • The term “high variety product production” meaning the production of products which in high flexibility/variation. I hope this clarifies.

“Hence, natural fiber reinforced polymer composite filaments seekhigh19attentionin the market.”: There is no connection to the last sentence and this one. So, why does author use “hence”?

  • The sentence has been revised.

The language used and the writing style are poor. Authors have to improve the language and the writing style should be more academic/research.

  • The manuscript has sent to English proofreading.

There are articles published earlier on 3D printing of kenaf based filaments unlike authors says in the text.

We are not able to locate any kenaf reinforced polymer composite filament for 3D printing purpose. Please do share us the information, we will study and include in the manuscript. 

Figure 2 directly copied from the source. Did authors take permission? It is common to adapt and redraw in author’s own way. The same for figure 3, 4, 5 …

  • The reuse permission required for tables and figures in our manuscript was obtained again, since multiple revisions has been done. The permission files were attached together in this correction.

References are not marked as per journal rules

  • The references have been updated as per journal guideline.

Only PLA polymer is mentioned. Isn’t there any other polymer that has been used?

  • It does have other polymer filaments used in this 3D printing technology such as polypropylene or polyamide. However, the reasons we choose PLA polymer are because it is the most developed and frequent used filament for 3D printing. Besides, the PLA filament is the only biodegradable polymer used in 3D printing currently.

Is table 2 even necessary for this review? Any relevance?

  • The table wish to show the similarity of kenaf fiber to other natural fibers. Hence, the kenaf fiber is workable as other natural fibers to embed in 3D printing filaments. Yet, it is not an essential table. Therefore, we decided to remove it.

Kindly see the below articles for further references

1) FDM 3D Printing of Polymers Containing Natural Fillers: A Review of their Mechanical Properties

2) Additive manufacturing (3D printing): A review of materials, methods, applications and challenges

3) A review of 3D and 4D printing of natural fibre biocomposites

4) 3D printing using plant-derived cellulose and its derivatives: A review

The current version has to be improved technically; and also the language must be improved

  • Thank you for the references, we are also well-aware on these impactful papers. We have carefully gone through once again and improve our paper. The language also improved in this revised manuscript.

Reviewer 3 Report

  1. The work clearly shows the state of the art of Kenaf fiber reinforcement PLA filaments for FDM 3D printing. This characteristic is observed in the text of the manuscript and in the citations presented to support the topic under study.
  2. The introduction progressively showed to the reader all the information necessary for the purpose of the study.
  3. The text is well written (definitions, interactions, process) and includes the objectives of the authors defined in the title.
  4. On the other hand, changes to the title of these sections and their texts could show the purpose of each section more directly.
  5. In addition, the "Challenges and future perspectives" section does not mainly show the challenges and future perspectives of potential of Kenaf fibers reinforcements in PLA oolymer Filaments for FDM. The authors use the section to talk about perspectives for the printing technique instead of focusing on the objective defined in the manuscript title.
  6. Text edits must be made for a second revision of the manuscript.

Author Response

The work clearly shows the state of the art of Kenaf fiber reinforcement PLA filaments for FDM 3D printing. This characteristic is observed in the text of the manuscript and in the citations presented to support the topic under study.

  • Thank you for your kind review of our manuscript. Your comments help us to reach a higher stage.

The introduction progressively showed to the reader all the information necessary for the purpose of the study.

  • Thank you for your compliment. We had revised some of our introduction section to make it better.

The text is well written (definitions, interactions, process) and includes the objectives of the authors defined in the title.

  • Thank you again for your time to handle our manuscript.

On the other hand, changes to the title of these sections and their texts could show the purpose of each section more directly.

  • We have revised the manuscript to make it in a better form.

In addition, the "Challenges and future perspectives" section does not mainly show the challenges and future perspectives of potential of Kenaf fibers reinforcements in PLA oolymer Filaments for FDM. The authors use the section to talk about perspectives for the printing technique instead of focusing on the objective defined in the manuscript title.

  • Thank you for your detailed complement, we do appreciate it.

Text edits must be made for a second revision of the manuscript.

  • The manuscript has sent to English proofreading.

Round 2

Reviewer 1 Report

Thank you for the changes done. The AM introduction still too long for this paper. Furthermore, I am stressing again that the Kenaf Reinforced PLA 3D printed parts review is small and the title of the paper cannot be the current one. Please change it to "natural fibers" for example.

Furthermore, please make clear in the paper that the reinforcement with synthetic fibers may be better in terms of the improved mechanical properties, but your reviews is only about natural fibers, etc. and these are mostly used in some fields where the synthetic ones cannot be used (please give proper examples).

In a scientific paper, you do not normally include figures from other articles, but some modified by the authors. I am not sure whether you have modified the images taken from other sources (even if you have permission to use them).

English language still needs corrections.

Author Response

Thank you for the changes done. The AM introduction still too long for this paper. Furthermore, I am stressing again that the Kenaf Reinforced PLA 3D printed parts review is small and the title of the paper cannot be the current one. Please change it to "natural fibers" for example.

  • The AM introduction has been shortened. We agreed with the view of the reviewer and changed the title to “natural fibers”.

Furthermore, please make clear in the paper that the reinforcement with synthetic fibers may be better in terms of the improved mechanical properties, but your reviews is only about natural fibers, etc. and these are mostly used in some fields where the synthetic ones cannot be used (please give proper examples).

  • A new section (4.2) has been added to discuss the comparison between natural and synthetic fibers.  

In a scientific paper, you do not normally include figures from other articles, but some modified by the authors. I am not sure whether you have modified the images taken from other sources (even if you have permission to use them).

  • We have obtained the permission from respective publishers to use the figure as it is.

English language still needs corrections.

  • We have checked the English language carefully and made sure it is acceptable. 

Reviewer 2 Report

The paper will be good overview for beginners in the field

Author Response

Thank you very much for your valuable time to review our manuscript